# Should Neurosurgeons Try to Preserve Non-Traditional Brain Networks? A Systematic Review of the Neuroscientific Evidence

**DOI:** 10.3390/jpm12040587

**Published:** 2022-04-06

**Authors:** Nicholas B. Dadario, Michael E. Sughrue

**Affiliations:** 1Robert Wood Johnson Medical School, Rutgers University, New Brunswick, NJ 08901, USA; nbd37@rwjms.rutgers.edu; 2Centre for Minimally Invasive Neurosurgery, Prince of Wales Private Hospital, Randwick, NSW 2031, Australia; 3Omniscient Neurotechnology, Sydney, NSW 2000, Australia

**Keywords:** connectome, neurosurgery, brain tumor, default mode network, central executive network, salience network, attention networks

## Abstract

The importance of large-scale brain networks in higher-order human functioning is well established in neuroscience, but has yet to deeply penetrate neurosurgical thinking due to concerns of clinical relevance. Here, we conducted the first systematic review examining the clinical importance of non-traditional, large-scale brain networks, including the default mode (DMN), central executive (CEN), salience (SN), dorsal attention (DAN), and ventral attention (VAN) networks. Studies which reported evidence of neurologic, cognitive, or emotional deficits in relation to damage or dysfunction in these networks were included. We screened 22,697 articles on PubMed, and 551 full-text articles were included and examined. Cognitive deficits were the most common symptom of network disturbances in varying amounts (36–56%), most frequently related to disruption of the DMN (n = 213) or some combination of DMN, CEN, and SN networks (n = 182). An increased proportion of motor symptoms was seen with CEN disruption (12%), and emotional (35%) or language/speech deficits (24%) with SN disruption. Disruption of the attention networks (VAN/DAN) with each other or the other networks mostly led to cognitive deficits (56%). A large body of evidence is available demonstrating the clinical importance of non-traditional, large-scale brain networks and suggests the need to preserve these networks is relevant for neurosurgical patients.

## 1. Introduction

Neurosurgeons have traditionally spent a great deal of time attempting to prevent injury in the so-called eloquent areas of the brain. Revolutionary advancements in neuroimaging technologies and intraoperative brain mapping technologies have expanded our ability to preserve motor and language functions in resective brain surgery while continuing to increase the extent of resection [1,2]. However, it is also clear that glioma patients still often present post-operatively with more subtle deficits in higher-order complex functions [3,4,5,6]. Complex neuro-behavioral functions, such as memory, attention, executive functioning, and emotion are commonly disrupted in brain tumor surgery, and can prevent patients from integrating back into society and the workforce [7]. Therefore, there has been a growing interest by some to reduce the cognitive footprint of supratentorial, intra-axial brain tumor surgery [8].

One reason we have previously been held back in the ability to optimize post-operative cognitive morbidity is because it has not been entirely clear exactly what we can and cannot do during surgery to avoid these problems. Although our study and subsequent understanding of brain anatomy responsible for language and motor functions has grown considerably in the last few decades, the neurosurgical community has generally maintained a less thorough familiarity with anatomy responsible for higher-order functions. One concept which has emerged from the field of neuroscience which may address this issue includes large-scale brain networks. These networks include reproducible areas that demonstrate highly synchronized activity based on specific functions or at rest, and are often measured with changes in blood oxygenation as a proxy for functional connectivity [9,10]. More recently, it has also been found that there are almost always large-scale white matter connections linking functionally connected regions within a structural network, illustrating: *regions that fire together are also wired together* [11,12,13,14,15]. Information on the structural and functional connectome has allowed for more precise maps and localization of complex patient symptomology that is not confined to isolated cortical regions [16,17].

Despite there being a large body of literature supporting this concept in the general field of neuroscience, the importance of the large-scale brain networks has still yet to deeply penetrate regular neurosurgical thinking. One reason for this stagnation may be because the clinical evidence and importance behind non-traditional, large-scale brain networks other than language and motor systems has yet to be systematically reviewed, and this may ultimately cause some to believe their presence may not be relevant for neurosurgery. Given that various neurological diseases and symptomology can often be better understood through consideration of their effects on networks [16], failure to consider brain network architecture hinders our successful movement as a community toward personalized neurosurgical treatments. Thus, here we attempt to address these concerns through a comprehensive review which aims to examine and outline the large body of available literature on this topic. Further, we raise the strong possibility that the features seen in other papers also apply to patients with brain tumors and may require further consideration moving forward. Through a brief systematic review, we outline and discuss the relevance of non-traditional eloquent areas in neurosurgery, specifically as it relates to damage or dysfunction in the large-scale brain networks: default mode network (DMN), central executive network (CEN), salience network (SN), dorsal attention network (DAN), and ventral attention network (VAN).

## 2. Materials and Methods

### 2.1. Search Strategy

A systematic review was conducted according to the Preferred Reporting Items for Systematic Reviews and Meta-Analysis (PRISMA) guidelines. An exhaustive screening process was completed using the electronic database PubMed. A search string was utilized with the following terms: “tractography OR fMRI” AND “memory OR language OR speech OR motor function OR attention OR depression OR neurologic deficit” AND “default mode OR central executive OR salience network OR sensorimotor OR language”. This search was conducted on 1 August 2021, for the period 2011–2021.

### 2.2. Selection Criteria

Articles were included which demonstrated the importance of the non-traditional, large-scale brain networks in relation to clinical symptoms or general disease states. Articles were included if they (1) mentioned fMRI or DTI techniques, (2) discussed the DMN, CEN, SN, DAN, VAN or related connections and network names, and (3) mentioned damage to or dysfunction in a network or related network alterations in association with neurologic, cognitive, or emotional deficits. Articles which clearly defined a disease state based on specific network changes in comparison with healthy controls were also examined. All article types, including reviews and conference abstracts, were included if the above criteria were met. Articles including only healthy subjects or traditionally known networks, such as language or motor networks, were excluded. Studies unable to provide an English text or which were strictly methodology focused were also excluded.

### 2.3. Screening and Data Extraction

A rigorous screening process was completed using EndNote and Rayyan (https://rayyan.qcri.org/, accessed on 29 September 2021). Title screening and abstract screening were completed by M.S. and the full-text review was completed by N.D. Included full-texts underwent a comprehensive qualitative review (N.D.) based on specific elements addressed, such as the network implicated, the type of network damage or dysfunction, the disease state, and the specific clinical symptoms implicated. Each article was also graded (M.D.) based on the class of evidence demonstrated.

## 3. Results

A total of 22,697 articles were screened and 1315 articles underwent full-text review (Figure 1). Ultimately, 551 full-text articles were included in final analyses and these are summarized below as well as included in the Appendix A.

### 3.1. Study Types

Studies included were generally of moderate study quality and demonstrated class III evidence (n = 524, 95%). Networks were mostly examined with functional connectivity analyses, such as resting-state fMRI to examine network dysfunction (n = 436, 79%) rather than in the context of network damage or structural integrity with DTI (n = 52, 9%). Combined structural-functional analyses were utilized in n = 63 (11%) studies to examine the relationship between network dysfunction in relation to its underlying structural network damage. Depression was the most common diagnosis (n = 124) described in the included studies. This was followed by Schizophrenia (n = 81), Parkinson’s disease (n = 64), Alzheimer’s disease (n = 46), stroke (n = 40), epilepsy (n = 29), and tumor cases (n = 21). These data are presented further and according to individual networks in Appendix A.

### 3.2. Networks Examined

The DMN (n = 213, 39%) represented the majority of articles identified in the current study. A number of works examined the effects of network disruptions in combination and therefore were further categorized into the most common combinations assessed. A triple network model of the DMN, CEN, and/or SN was the next most commonly studied network (n = 182, 33%). Similarly, the attention networks (VAN and DAN) were only examined in isolation in one study [18], and instead were mostly studied in combination with the other brain networks or each other (n = 93, 17%). Few studies were identified on the CEN (n = 25, 5%) in the current study. However, a number of studies were excluded that discussed the superior longitudinal fasciculus, a major fiber bundle linking the CEN, but were not clearly discussed in reference to the CEN. Differences in the diagnosis frequency per network are further demonstrated in Appendix A.

### 3.3. Cognitive, Emotional, and Neurologic Deficits

Cognitive deficits (n = 226, 41%), such as in attentional processing, memory, and executive functioning, were the most commonly identified outcomes of network disturbances. This was followed by emotional processing deficits in 26% (n = 143) of studies, mostly in the context of affective traits in depression and/or anxiety. A number of studies were identified which demonstrated deficits in more than one outcome, such as emotion and cognition (6%), motor and cognition (5%), motor alone (4%), language or speech (4%), or many additional deficits together (7%).

Deficits were also analyzed based on their frequency in each network or network combination (Figure 2). Cognitive deficits demonstrated the largest proportion of all network or network combination deficits in varying amounts (36–56%), except for the SN which mostly consisted of emotional processing deficits (35%) following network disturbance. The CEN and SN demonstrated the most diverse range of symptoms according to frequency, with a noted increase in motor symptoms with the CEN (12%) and language processing or speech deficits with the SN (24%). Disruption of the attention networks (VAN/DAN) in combination with each other or the other networks most commonly led to cognitive deficits (56%).

## 4. Discussion

In this paper, we have performed the first systematic review that examines and outlines the literature on major non-traditional, large-scale brain networks as it relates to neurosurgery. A large amount of plausible evidence was identified suggesting that disturbances in non-traditional networks can result in severe neurological, cognitive, or emotional deficits. Although it remains unclear if or what specific damage to a brain network causes a specific neurologic deficit, the available literature seems to suggest that at least some deficits will occur when key network regions or their interconnecting fibers are disrupted. However, to date, this information has only been briefly discussed and has yet to be systematically reviewed in a way which can demonstrate their clinical importance for regular neurosurgical thinking. Given these neuroanatomic substrates are commonly encountered during resective brain surgery, below we attempt to briefly characterize their relative importance based on the previous literature and current neurosurgical practices, and also elucidate how features seen in many different papers can be especially relevant to brain tumor patients. Importantly, as a first step, the current review provides a broad overview of the likely clinical importance of these networks and how incorporating their presence in regular neurosurgical thinking may provide us a more nuanced, personalized approach to connectome-based neurosurgery moving forward.

### 4.1. Who Are the Big Five Non-Traditional Brain Networks?

Brain networks have been found to be central to the organizing principle of the structural and functional brain connectome. Although some networks may be new to many in neurosurgery, it is important to note that the default mode, central executive, salience, dorsal attention, and ventral attention networks are far from fringe in the broader neuroscientific community and have been well-documented over the past 20 years. These networks are made up of highly synchronized cortical regions and the interconnecting white matter bundles between these regions. As it relates to resective brain surgery, the structural integrity of large-scale networks is paramount for patients to maintain a level of functional capacity and metabolic efficiency necessary to subserve complex human functions (Table 1).

Three canonical resting-state networks which sit at the top of the network hierarchy and define an axis by which the other networks align for complex neuro-behavioral functions are the DMN, CEN, and SN (Figure 3). Although only briefly discussed previously [8], results from the current work strongly suggest that disruption of these networks can commonly lead to cognitive and emotional deficits, which are also thought to underlie a number of neuro-psychiatric illnesses such as depression [19]. In contrast, the attention networks chiefly work in tangent with other large-scale networks for top-down and bottom-up processing of stimuli and to re-orient attention based on internal and external motivations (Figure 4) [11,20,21]. Unsurprisingly, disruption of the VAN or DAN when in combination with the other higher-order networks most commonly leads to impaired cognitive processes, such as in spatial neglect [22] and impaired memory [23]. Given that these five networks seem to be a common feature that explains much of the human functioning we understand in neuroscience and are commonly encountered during resective brain surgery, it is likely safe to say they are more important than previously considered, and that whether they need to be preserved is a question worth asking.

### 4.2. Evidence for Avoiding Damage to Brain Networks in Brain Tumor Patients

There is no randomized controlled trial showing that not destroying one of these five non-traditional, large-scale networks in a surgery leads to improved neurological outcomes. Despite this limitation, the evidence that damage or dysfunction in these networks causes a cognitive, neurological, or emotional disturbance is relatively vast and should be considered as such. Most of the previous literature has focused on network disturbances in the context of neuro-psychiatric illnesses, where a clear neuroanatomic substrate or lesion is not readily identifiable but the pathophysiology clearly involves widespread dysfunction across numerous spatially distant regions involved in interacting networks [26,27]. However, psychiatric symptoms can often be the primary and only symptoms of brain tumors as well due to similar network disturbances which alter neural connectivity [28,29] and this has only been recently acknowledged. Therefore, despite some differences between the mechanisms of network disruption, decisions during resective brain surgery may also benefit from consideration of these five non-traditional networks as there are clear deficits and widespread effects associated with their disruption when key network regions or fibers are cut [30,31].

Based on this review, a significant proportion of studies demonstrated cognitive and emotional deficits in the context of multiple, interacting brain networks. This is unsurprising given most higher-order functions are not localized to one specific region of brain tissue, but instead rely on numerous dynamic interactions amongst higher-order networks such as the DMN, CEN, and SN (Figure 5). The importance of these cognitive control networks have been increasingly demonstrated through various lines of evidence [19,30,31,32], including brain tumor, stroke, and psychiatric patients. Together, these data have suggested that disruption in these networks, often following damage in the medial frontal lobe, may cause cognitive and psychomotor disturbances related to an abnormal allocation of cognitive resources between networks [30,31,33]. An example of this can be seen with butterfly gliomas which often envelop non-traditional networks such as the DMN and/or SN, and when damaged in surgery can result in post-operative abulia and akinetic mutism (Figure 5) [33].

Ultimately, without consideration of the large-scale, non-traditional brain networks, we are left with the traditional localizationist view: *if we cut across any of these networks, then no deficit will occur*. Alternatively, the available body of evidence found in the current review suggests that if you cut across networks such as the DMN, there may be at least some consequences in some patients, if not most. Simultaneously, it must be noted that these results do not suggest we cannot remove tumors involved in major brain networks. Instead, the current report suggests that considering their connectomic architecture and presence in our thinking during brain tumor surgery may provide additional information of prognostic value for the operating neurosurgeon [34,35].

### 4.3. Brain Network Maps vs. Other Surgical Adjuncts: Mutually Exclusive?

Decisions made during surgery to preserve higher-order functions have previously been made based on incomplete anatomic information that also ignores inter-individual differences in brain structural–functional relationships, often leading to significant consequences [3,4,5,6,36]. Brain networks provide clear maps which can define specific onco-functional resection boundaries, or at least offer additional information to the operating neurosurgeon to make more informed decisions during surgery while causing fewer deficits [8]. However, a common concern expressed by many neurosurgeons is that the consideration of brain networks may not be necessary if one already employs awake surgery techniques to identify eloquent cortices. Nevertheless, it is important to note that raising such concerns may be misleading, as awake surgery and brain network mapping are not mutually exclusive. In fact, these two surgical tools are complementary as awake surgery relies on a strong anatomical understanding of where specific regions are located before surgery [33,37,38]. Network maps provide the best available tool to date to make this anatomy known, and likely provide additional benefits for awake surgeons regardless of previous experience due to there being more details about brain anatomy between individual patients that can be estimated before and during surgery.

In the pre-operative period, brain network maps can identify the relative risks of specific tumors [39,40], outline the functional boundaries of resection [31,33], and clarify subtle differences in trajectory according to surrounding networks (Figure 6) [41,42]. However, unlike brain network maps, awake surgery is not appropriate in many cases [43]. Glioblastoma (GBM) patients often present with substantial pre-operative neurological deficits and therefore are often not suitable awake craniotomy candidates due to an increased risk of adverse events. Unfortunately, maintaining a binary view of “awake or nothing” may make these deficits a self-fulfilling prophesy, where we unknowingly destroy their network which otherwise may just be compressed or edematous.

Furthermore, it is important to note the relative difficulty in mapping higher cognitive functions during awake surgery. For instance, it hard to consider what test protects the default mode network function, or, if you find a positive site for verbal memory, what you should do to keep that area connected to the rest of the system. Although there have been some improvements in the ability to map higher cognitive functions using awake surgery, it remains difficult to save a brain function when the complete anatomy underlying that function is unknown [37].

### 4.4. Do You Put Patient Survival at Risk to Preserve the Network?

An important point to address is that using brain network maps does not inherently require decisions to be made in specific patients between destroying a network core or decreasing patient survival by leaving substantially more tumor. Such a question is a drastic oversimplification, and is rarely presented when utilizing brain network maps according to our experience. Instead, a more appropriate question brain networks allow us to pose is: *is it worth leaving the last 1–5% of tumor to prevent a specific patient from having serious cognitive problems after this surgery?*

In glioma surgery, comparing a 98% resection with good neurologic function with a 100% “complete resection” with somewhat worse function, given that neither is generally curative for most gliomas, is a philosophical question. Although many would argue that an additional 2% extent of resection may only provide negligible additional benefits that do not justify a trade-off in neurologic function, brain network maps allow us to better consider the risks associated with these decisions [44]. By defining the tumor and personalized surgical plan against clear network boundaries marked by white matter bundles with known functional significance, we may be able to make more informed decisions during surgery on when to and when not to extend the rate of resection according to pre-defined patient onco-functional goals on an individualized patient basis [45,46]. We may ultimately choose to ignore this network information due to the inability to preserve a network completely enveloped in tumor, or rather use it to achieve a supramaximal resection up to key network structures to preserve significant functions (Figure 6) [41]. Regardless, network information may allow us to better consider the risks and benefits of these surgical decisions, the risks of certain tumors, or even when to consider alternative radiotherapeutic strategies that have continued to advance in precision.

### 4.5. What We Still Do Not Know

Despite the clear benefits posed with improved knowledge of the brain connectome architecture in neurosurgical patients, there remain a number of areas requiring further research to better leverage this information.

Importantly, if a network is invaded by a tumor, it is still not completely clear how we are to understand if that network is salvageable or not. Leaving residual tumor in a network may not ultimately be worth it if the tumor is disrupting a core network region, and at other times attempting to save a network may be far from surgically feasible or practical. Furthermore, it is unclear if it possible to remove a tumor and save the underlying network, as we could do for an acoustic neuroma, for example [47]. However, in many cases, brain tractography allows us to consider adjusting the surgical trajectory by just a few millimeters to avoid the network, and still completely remove the tumor and save the network (Figure 6) [48].

*How do we know if the other side of a network or another region can compensate for losing part of the network*? This is a great question for connectomics in the future as we understand a great portion of the human brain maintains a certain degree of redundancy [49]. A large amount of this redundancy likely originates from the contralateral hemisphere, given most functions are processed bi-hemispherically. When operating in the SMA with a tumor invading the salience network, preservation of this network’s transcallosal fibers (“crossed FAT”) may allow patient recovery from SMA syndrome [50]. However, in our experience, the other side of a network does not always compensate well for losses in core parts of a network, such as with the default mode network, and this should be considered prospectively. Improved statistical modeling techniques of the brain connectome are underway to develop clinically applicable metrics which can estimate these neurological features; however, further clinical study is necessary [49].

Lastly, it remains unclear to what degree small distant parts of a network matter if they are not part of its core. For instance, the CEN is a three-part network with parcellation clusters in the frontopolar region, posterior DLPFC, and inferior parietal lobe (Figure 3). Damaging the middle aspect of this network and key fibers linking all other frontal and parietal parcellations likely disrupts the entire network’s function. However, the ability to sacrifice a portion of the frontal cluster or parcellations in the periphery without disrupting overall network functioning remains unclear, and we have seen cases where disconnecting these regions from the network in surgery creates mild clinical deficits.

These are just some of the important questions that should be the subject of future research efforts. It is important to know, however, that the reason these answers remain unknown is because up until this point, we did not have a common nomenclature for studying these networks, nor sufficient tools for having meaningful discussions and developing technique refinements in this area. Moving forward, consideration of the non-traditional brain networks in regular neurosurgical thinking provides additional opportunities to optimize the patient onco-functional balance following resective brain surgery.

## 5. Conclusions

What we can conclude is that there is a substantial body of plausible evidence suggesting that damage or dysfunction in the non-traditional, large-scale brain networks may cause severe neurological, cognitive, or emotional deficits. The default mode, central executive, salience, dorsal attention, and ventral attention networks explain much of the complex human functioning currently understood in the neuroscientific community and therefore whether there is a need to preserve these non-traditional “eloquent” regions is an important question that should be incorporated into regular neurosurgical thinking moving forward. Increased use of rigorous pre- and post-operative neurocognitive assessments remains a priority moving forward, and these outcomes should be linked with data on network disturbances in larger trials in order to advance our understanding of and subsequent ability to optimize cognitive outcomes following brain surgery.

## Figures and Tables

**Figure 1 jpm-12-00587-f001:**
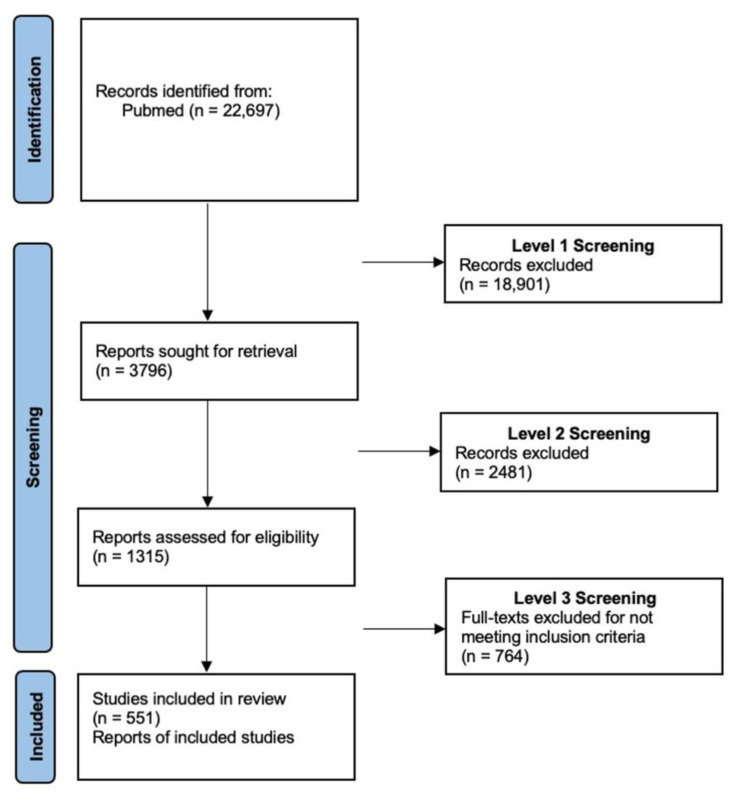
Preferred Reporting Items for Systematic Reviews and Meta-Analysis (PRISMA) Checklist.

**Figure 2 jpm-12-00587-f002:**
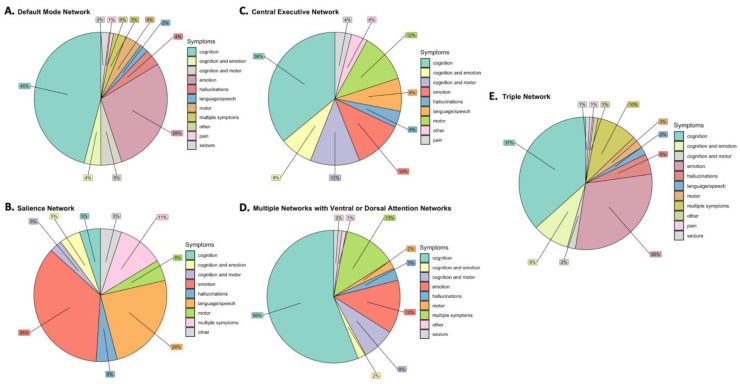
Deficits Associated with Network Disturbances. Deficits are presented based on relative frequency for the (**A**) Default Mode Network (DMN), (**B**) Salience Network (SN), (**C**) Central Executive Network (CEN), (**D**) Ventral or Dorsal Attention Network in combination with each other or the other networks, and (**E**) Triple network combination of the DMN, SN, and CEN. Different colors represent different deficits as shown in the figure legend.

**Figure 3 jpm-12-00587-f003:**
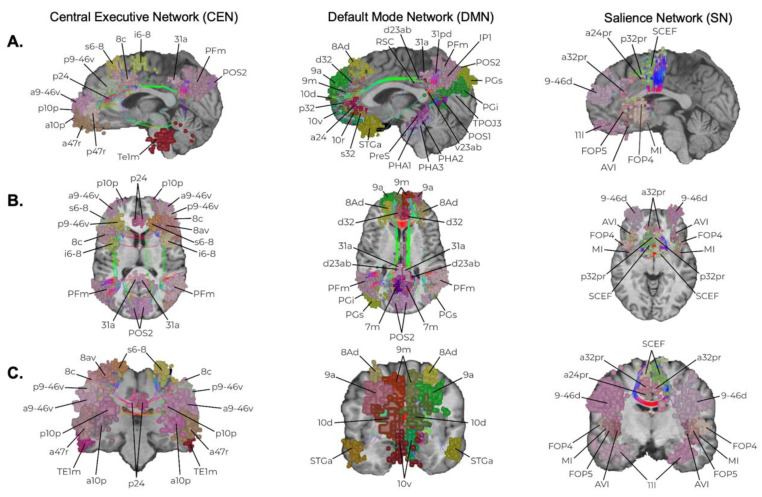
Cognitive Control Networks. The anatomic locations of the Default Mode Network (DMN), Salience Network (SN), and Central Executive Network (CEN) are presented in sagittal (**A**), axial (**B**), and coronal slices (**C**). Individual parcellations are indicated according to the established Glasser Parcellation Scheme of the Human Connectome Project [24] based on network affiliation [25].

**Figure 4 jpm-12-00587-f004:**
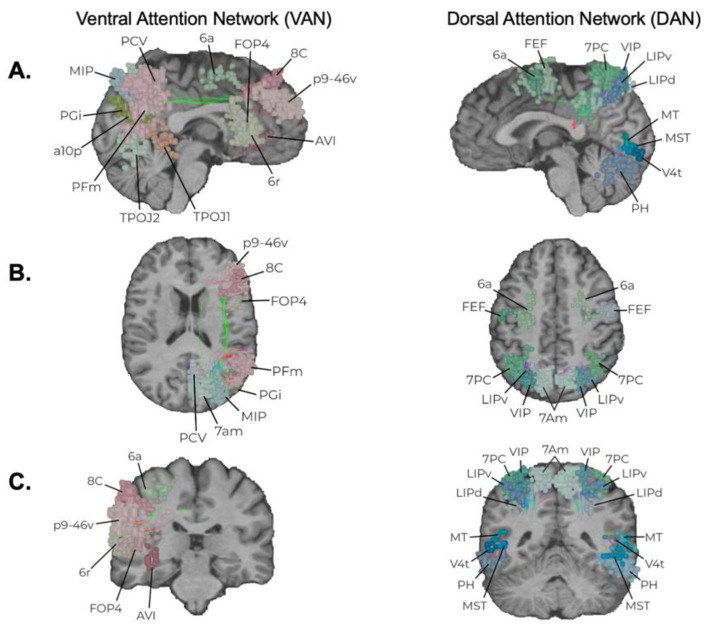
Attention Networks. The anatomic locations of the Dorsal Attention Network (DAN) and Ventral Attention Network (VAN) are presented in sagittal (**A**), axial (**B**), and coronal slices (**C**). Individual parcellations are indicated according to the established Glasser Parcellation Scheme of the Human Connectome Project [24] based on network affiliation [25].

**Figure 5 jpm-12-00587-f005:**
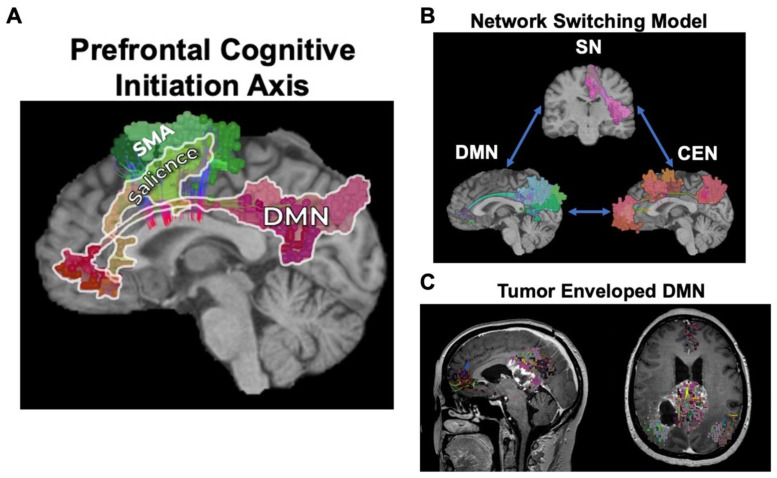
Prefrontal Cognitive Control Networks. (**A**) The Prefrontal Cognitive Initiation “Axis” is demonstrated where the DMN, connected via the cingulum, and the SN, connected via the frontal aslant tract, form a strip across the medial frontal lobe up until the supplementary motor area (SMA). (**B**) The network switching model, or triple network, is represented. The salience network (SN) controls the allocation of cognitive resources between internally focused (DMN) and externally focused (CEN) network functions. (**C**) A posterior butterfly glioma of the splenium is seen enveloping the DMN and demonstrates the risk of butterfly glioma surgery in this area on DMN integrity.

**Figure 6 jpm-12-00587-f006:**
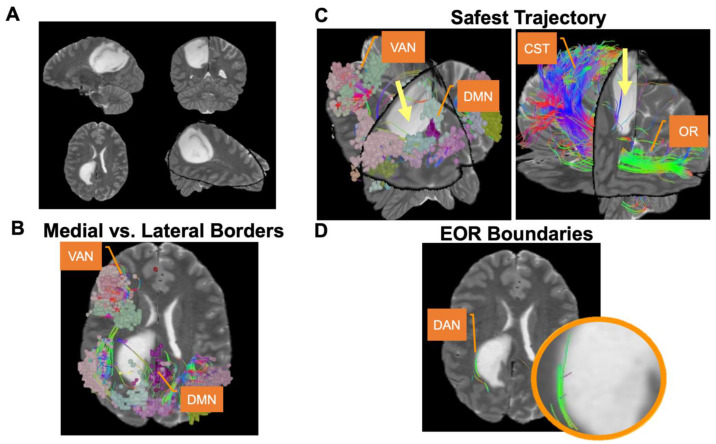
Use of Brain Network Maps for Glioma Surgery. Consideration of the non-traditional, large-scale brain networks is utilized in a case example of a patient with a right parietal anaplastic oligodendroglioma. (**A**) T2-weighted magnetic resonance imaging is shown in sagittal (top left), coronal (top right), axial (bottom left), and three-dimensional (bottom right) views. (**B**) The tumor is preoperatively defined against the ventral attention network (VAN) as the lateral boundary and the default mode network (DMN) as the medial boundary. (**C**) A trajectory is chosen against brain networks so as to minimize unnecessary network disturbances on the entry to the tumor. (**D**) A zoomed-in view demonstrates the tumor margin pressed against the superior longitudinal fasciculus (SLF) of the dorsal attention network (DAN).

**Table 1 jpm-12-00587-t001:** Five non-traditional, large-scale brain networks.

	Central Executive Network (CEN)	Default Mode Network (DMN)	Salience Network (SN)	Ventral Attention Network (VAN)	Dorsal Attention Network (DAN)
Core of Network	Lateral frontopolar region, posterior DLPFC, supramarginal gyrus	Anterior cingulate, posterior cingulate, and lateral parietal network	Anterior insula and frontal operculum, middle cingulate; middle aspect of DLPFC	Middle and inferior frontal gyrus, anterior insula, superior and inferior parietal lobules, and temporo-parietal junction(Right hemisphere)	Frontal eye fields, intraparietal sulcus, superior parietal lobule, and visual cortex
Key long-range fiber	Superior longitudinal fasciculus (SLF)	Cingulum	Frontal aslant tract (FAT)	Superior longitudinal fasciculus (SLF)	Superior longitudinal fasciculus (SLF)
Key Functions	External, active thought Executive function Working memory	Internal, passive thought Theory of Mind Imagination Episodic Memory	Allocation of cognitive resources DMN/CEN switching Saliency detection Emotional regulation	Bottom-up processing with preconscious stimuli Change in attention with sensory input	Top-down, active processing Sustained direct attention

Abbreviations: DLPFC, dorsolateral prefrontal cortex.

## Data Availability

All data utilized in the current study have been provided in the Appendix A.

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
