# Peer review of "Should Neurosurgeons Try to Preserve Non-Traditional Brain Networks? A Systematic Review of the Neuroscientific Evidence"

_jpm, 2022, doi:10.3390/jpm12040587_

Round 1

Reviewer 1 Report

In this article, the authors systematically organized the effect of large-brain networks and their effects on functional outcomes. The methods described in the article seem reasonable and the summarized results are potentially helpful for future research and surgeries. The authors also discussed the potential influence of their results in great depth. I would recommend the publication of this manuscript.

Author Response

Revisions for Brain Networks Review Manuscript

In this article, the authors systematically organized the effect of large-brain networks and their effects on functional outcomes. The methods described in the article seem reasonable and the summarized results are potentially helpful for future research and surgeries. The authors also discussed the potential influence of their results in great depth. I would recommend the publication of this manuscript.

The authors of the current manuscript would like to thank reviewer 1 for their careful consideration and review of our manuscript. This topic is one that we feel strongly about and we hope that this manuscript provides a body of meaningful data for your readership should it be accepted in order to further the discussion on this topic in the neurosurgical community.

The manuscript has been proof checked to ensure adequate publication in accordance with the reviewer’s recommendation. Should the reviewer request further edits to the current manuscript, the authors would be happy to provide further revisions upon the reviewers’ requests.

Reviewer 2 Report

The authors propose their systematic review on the non-traditional brain networks, their significance and implications to neurosurgery.

The review in general is well carried out, and the results are clearly presented through several figures.

From the neurosurgical point of view, these zones are usually considered silent and the resection through is advised. Furthermore, when these deficits appear, in a traditional fashion, they are attributed to anaesthesia or anti-epileptic drugs related complications, and are rarely considered as surgery related. Nevertheless, these underestimated complications may lead to significant compromise of a QoL.

This review and reviewed papers emphasised the importance of cognitive assessment pre- and post-operatively, however, the authors should mention this within conclusions, as it is a first step in introduction of advanced neuroscientific knowledge in neurosurgical practice.

On the other hand, recent development of surgical technique have emphasised on the avoidance of all kinds of deficits, the surgery might lead to, and better brain tissue preservation with the use of various technological advances.

The manuscript is very well thought and written. 

Author Response

Revisions for Brain Networks Review Manuscript

The authors of the current manuscript would like to thank reviewer 2 for their careful consideration and review of our manuscript. We have accepted the reviewer’s concerns, and have replied below appropriately to their comments as well as made all updates to the current manuscript with tracked changes in order to demonstrate all appropriate updates made by our team separate from what has previously been recorded.

In addition to addressing the reviewer’s suggestions, the manuscript has been further proof checked for any grammatical errors. Should the reviewer request further edits to the current manuscript, the authors would be happy to provide further revisions upon the reviewers’ requests.

Reviewer 2:

The authors propose their systematic review on the non-traditional brain networks, their significance and implications to neurosurgery.

The review in general is well carried out, and the results are clearly presented through several figures.

From the neurosurgical point of view, these zones are usually considered silent and the resection through is advised. Furthermore, when these deficits appear, in a traditional fashion, they are attributed to anaesthesia or anti-epileptic drugs related complications, and are rarely considered as surgery related. Nevertheless, these underestimated complications may lead to significant compromise of a QoL.

This review and reviewed papers emphasised the importance of cognitive assessment pre- and post-operatively, however, the authors should mention this within conclusions, as it is a first step in introduction of advanced neuroscientific knowledge in neurosurgical practice.

On the other hand, recent development of surgical technique have emphasised on the avoidance of all kinds of deficits, the surgery might lead to, and better brain tissue preservation with the use of various technological advances.

The manuscript is very well thought and written. 

  • We sincerely thank the reviewer for their kind remarks. We agree with the reviewer’s comments as this is a topic we feel strongly about and we hope that this manuscript provides a body of meaningful data for your readership should it be accepted in order to further the discussion on this topic in the neurosurgical community.
  • In regards to the reviewers suggestion in the last comment, we have since went back and made these edits as the reviewer suggested. This point has been strongly emphasized throughout the manuscript so we agree with the reviewer that also emphasizing it in the conclusion is important to ultimately deliver our message.
    • Edited conclusions lines 474-477 (edited text is in purple)
    • What we can conclude is that there is a substantial body of plausible evidence suggesting that damage or dysfunction in the non-traditional, large-scale brain networks may cause severe neurological, cognitive, or emotional deficits. The default mode, central executive, salience, dorsal attention and ventral attention networks explain much of the complex human functioning currently understood in the neuroscientific community and therefore the need to preserve these non-traditional “eloquent” regions is an important question that should be incorporated into regular neurosurgical thinking moving forward. Increased use of rigorous pre- and post-operative neurocognitive assessments remains a priority moving forward, and these outcomes should be linked with data on network disturbances in larger trials in order to advance our understanding of and subsequent ability to optimize cognitive outcomes following brain surgery.